# Clonal dynamics towards the development of venetoclax resistance in chronic lymphocytic leukemia

Carmen D. Herling[1], Nima Abedpour[2,3], Jonathan Weiss[1], Anna Schmitt[1,3,4], Ron Daniel Jachimowicz[1,3,4], Olaf Merkel[1,4], Maria Cartolano[2,3], Sebastian Oberbeck[1,3,4,5], Petra Mayer[1,3,4,5], Valeska Berg[1,4], Daniel Thomalla[1,4], Nadine Kutsch[1], Marius Stiefelhagen[1], Paula Cramer[1], Clemens-Martin Wendtner[6], Thorsten Persigehl[7], Andreas Saleh[8], Janine Altmüller[3,9], Peter Nürnberg[3,4,9], Christian Pallasch[1,3,4], Viktor Achter[10], Ulrich Lang[10,11], Barbara Eichhorst[1], Roberta Castiglione[12], Stephan C. Schäfer[12], Reinhard Büttner[12], Karl-Anton Kreuzer[1], Hans Christian Reinhardt[1,3,4], Michael Hallek[1,3,4], Lukas P. Frenzel[1,4] & Martin Peifer [1,4]

Deciphering the evolution of cancer cells under therapeutic pressure is a crucial step to understand the mechanisms that lead to treatment resistance. To this end, we analyzed whole-exome sequencing data of eight chronic lymphocytic leukemia (CLL) patients that developed resistance upon BCL2-inhibition by venetoclax. Here, we report recurrent mutations in *BTG1* (2 patients) and homozygous deletions affecting *CDKN2A/B* (3 patients) that developed during treatment, as well as a mutation in *BRAF* and a high-level focal amplification of *CD274* (*PD-L1*) that might pinpoint molecular aberrations offering structures for further therapeutic interventions.

[1] Department of Internal Medicine I, Center of Integrated Oncology Cologne-Bonn, University of Cologne, 50937 Cologne, Germany. [2] Department of Translational Genomics, Center of Integrated Oncology Cologne-Bonn, Medical Faculty, University of Cologne, 50931 Cologne, Germany. [3] Center for Molecular Medicine Cologne (CMMC), University of Cologne, 50931 Cologne, Germany. [4] Cologne Excellence Cluster on Cellular Stress Response in Aging-Associated Diseases (CECAD), University of Cologne, 50931 Cologne, Germany. [5] Laboratory of Lymphocyte Signaling and Oncoproteom, University of Cologne, 50931 Cologne, Germany. [6] Department of Hematology, Oncology, Immunology, Palliative Care, Infectious Diseases and Tropical Medicine, Klinikum Schwabing, 80804 Munich, Germany. [7] Department of Radiology, Cologne University Hospital, 50937 Cologne, Germany. [8] Department of Diagnostic and Interventional Radiology and Pediatric Radiology, Städtisches Klinikum München Schwabing, 80804 Munich, Germany. [9] Cologne Center for Genomics (CCG), University of Cologne, 50931 Cologne, Germany. [10] Computing Center, University of Cologne, 50931 Cologne, Germany. [11] Department of Informatics, University of Cologne, 50931 Cologne, Germany. [12] Department of Pathology, University of Cologne, 50937 Cologne, Germany. Carmen D. Herling, Nima Abedpour, Lukas P. Frenzel, and Martin Peifer contributed equally to this work. Correspondence and requests for materials should be addressed to M.P. (email: mpeifer@uni-koeln.de)

Chronic lymphocytic leukemia (CLL) is the most common leukemia in the western world with a diverse clinical course and a substantial degree of inter- and intra-patient heterogeneity[1–8]. Ongoing clonal evolution is sought to be a key mechanism for the development of treatment-resistant or -refractory CLL[3,4,9] and is, therefore, limiting treatment success and duration. Data from whole-exome or genome sequencing can be used to reconstruct the clonal evolution of cancer specimens and has led to a deeper understanding of underlying principles[10–13]. In CLL, studies of clonal evolution under treatment with the Bruton tyrosine kinase (BTK) inhibitor ibrutinib have provided novel insights into changes of the clonal architecture towards treatment resistance[3,14].

Recently, treatment with the BCL2-inhibitor venetoclax has demonstrated substantial efficacy, even in high-risk relapsed and chemotherapy-refractory CLL patients with alterations in TP53[2,15,16]. This group of patients has a particularly poor prognosis[2]. In contrast to ibrutinib, genetic causes underlying resistance to the treatment with venetoclax have not been determined, so far. To this end, we conducted whole-exome sequencing and methylation profiling of serial CLL samples obtained from eight individuals before initiation of venetoclax treatment and at the manifestation of resistance. From these data, we inferred the clonal evolution under venetoclax therapy and identified recurrent genome alterations that developed during treatment.

## Results

**Samples and clinical data.** All patients were previously treated (1–8 previous lines of treatment) and harbored either genomic losses (del(17p)) or protein-damaging point mutations in TP53 in all samples. The median time between the initiation of venetoclax treatment to clinical progression/relapse was 15.4 months (Supplementary Table 1). Partial remission as best response to venetoclax was achieved in six patients (Fig. 1a, Supplementary Fig. 1). Patient C586 presented a stable disease with disappearance of lymphocytosis, but only a reduction of nodal mass by 41% (Fig. 1a), and patient C548 showed a stable disease, due to persisting splenomegaly. Time to progressive disease ranged between 4 and 22 months. Half of the patients developed a Richter's transformation (RT) during venetoclax treatment (Fig. 1b, Supplementary Table 1), histologically presenting as diffuse large B-cell lymphoma.

**Accumulation of genome alterations during treatment.** On average, we detected a total of 25.5 exonic mutations (including synonymous and non-synonymous point mutations, insertions, and deletions) prior to venetoclax therapy, in accordance with previous CLL exome-sequencing efforts[3]. As expected for ongoing evolutionary processes, copy number changes and the number of point mutations increased during venetoclax therapy (Fig. 1b). Remarkably, for patient C651 we did not detect any copy number change before venetoclax treatment, but 12.5% of the genome showed losses or gains at the occurrence of venetoclax resistance (Fig. 1b). At time of relapse, only lymph node specimens with a lower purity were available for most patients, in contrast to the generally pure pre-treatment samples derived from peripheral blood (Fig. 1b, Supplementary Table 2). This hampers a robust assessment of treatment-specific changes in the methylation profiles, since most of the variability seen in the methylation patterns within each patient is likely due to differences in the cell type composition of the compartments analyzed (Supplementary Fig. 2).

**Alterations in cancer-related genes.** We next selected somatic genome alterations in cancer-related genes that showed clonal dynamics during venetoclax therapy. Recurrent non-synonymous mutations were seen in: TP53, NOTCH1, BTG1, whereas BRAF, SF3B1, RB1, BIRC3, and MLL3 were non-synonymously affected in single patients only (Fig. 1b). These mutations were validated by either dideoxy sequencing or digital droplet PCR (except for TP53 mutations that were previously assessed at study inclusion)[16] (Supplementary Fig. 3). Note that the mutation in BIRC3 could not be validated, due to a lack of genomic material after whole-exome sequencing. Allelic fractions between digital droplet PCR and whole-exome sequencing were highly comparable for single-nucleotide mutations (Supplementary Fig. 3a). Furthermore, we observed homozygous deletions of CDKN2A/B and a high-level focal amplification containing CD274 encoding for the immune-checkpoint ligand PD-L1 (Fig. 1b). Copy numbers from exome sequencing were validated by methylation arrays (Supplementary Fig. 4).

In line with previous findings[15,16] patients responded to venetoclax therapy, even if TP53 was initially mutated in a bi-allelic fashion (5/8 patients). Two patients showed genome alterations that might qualify for further therapeutic options after, or in combination with venetoclax therapy: (1) patient C548 harbored a BRAF (p.K601E) mutation that was shown to be oncogenic and can be targeted by, e.g., MEK inhibitors[17], and (2) the CD274 amplification, which was paralleled by high CD274 protein expression levels and a prominent infiltrate of CD3-positive T cells (Fig. 1c) may be susceptible to immune-checkpoint blockade[18] in patient C811. High-level and focal amplifications of CD274 have been described in a variety of human cancer entities[19,20], but so far not in CLL. In contrast, a pooled analysis of two major CLL-sequencing studies[3,8] revealed that mutations in BRAF occur at a frequency of 3.8% ($n = 559$) in treatment-naive samples, and can, therefore, be considered as a potential driver alteration in CLL.

**Evolved recurrent somatic alterations.** Recurrent genomic changes that evolved during venetoclax treatment were homozygous deletions affecting CDKN2A/B in three patients (C548, C577, C586) and BTG1 missense mutations in two cases (C577: p.Q36H; C789: p.E46K). BTG1 has been shown to counteract cell proliferation and to be regulated downstream of BCL2 and CDKN2A/B (p16$^{Ink4a}$/p14$^{Arf}$)[21,22]. Thus, aside the abrogation of cell cycle control by loss of CDKN2A/B, damaging mutations in BTG1 may provide a survival advantage to CLL cells under targeted BCL2-inhibition. Non-synonymous mutations affecting BTG1 have been rarely detected in untreated CLL patients (only in 3 of 559 samples)[3,8]. Therefore, the probability that the BTG1 mutations developed spontaneously and are not due to a selection process by the venetoclax treatment is $7.8 \times 10^{-4}$. To assess the frequency of homozygous CDKN2A/B deletions, we queried 125 samples for which copy number data were publically available[8]. This analysis revealed that homozygous deletions of CDKN2A/B were not detected in these treatment-naive CLL samples. However, homozygous loss of CDKN2A/B has been associated with CLL cases that have undergone RT[23,24]. Intriguingly, two out of the three cases that developed homozygous CDKN2A/B deletions in our series had undergone RT at relapse (Fig. 1b). Furthermore, all patients with homozygous deletions of CDKN2A/B also harbored other cancer-related mutations (Fig. 1b).

**Clonal dynamics under venetoclax treatment.** To gain insight into the clonal evolution towards therapy resistance, we inferred subclonal populations and reconstructed phylogenetic trees (see Methods). Intriguingly, we observed a wide spectrum of evolutionary dynamics, including linear, divergent, and convergent

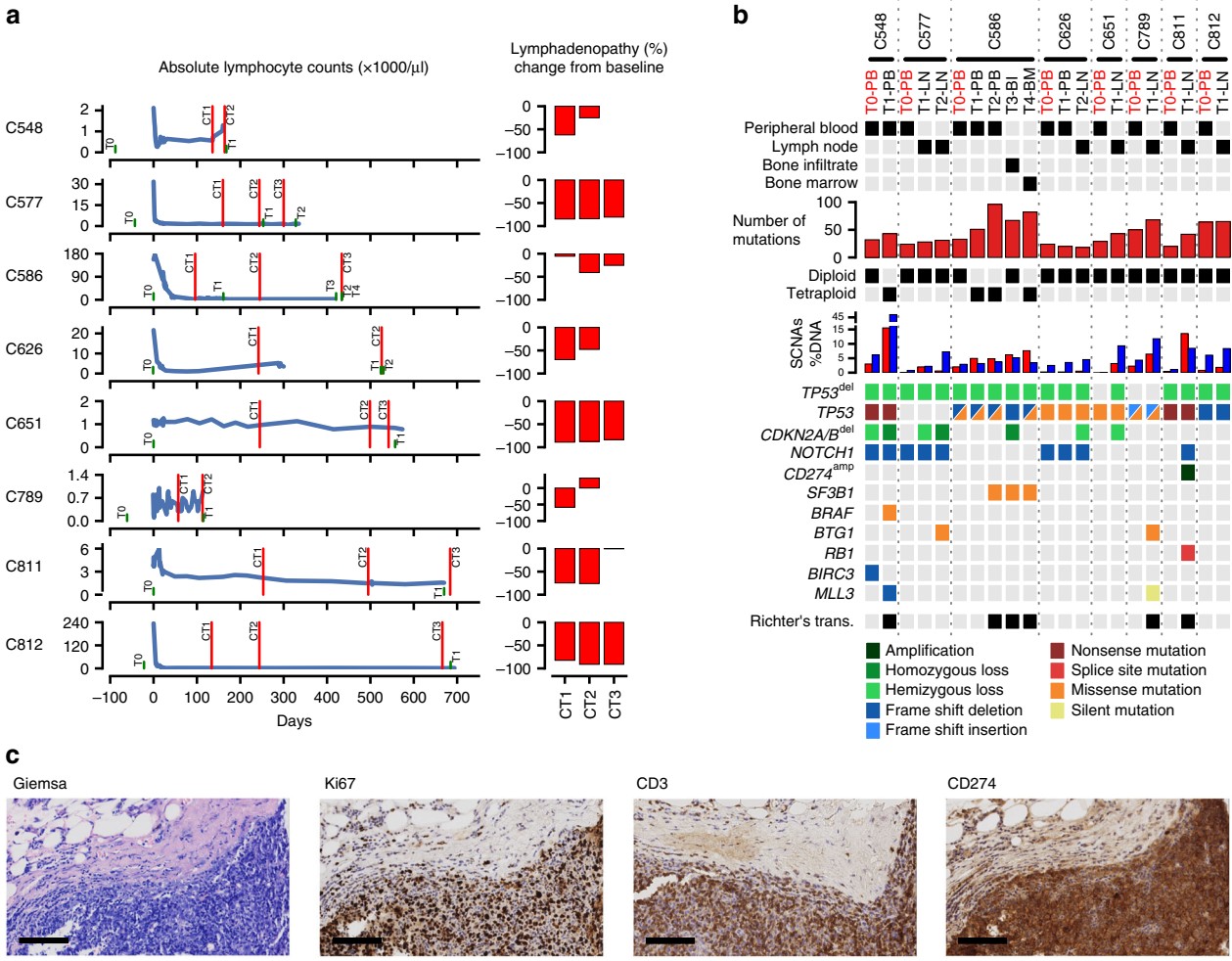

**Fig. 1** Patient and their related matched pre-treatment and relapse samples characteristics. **a** Absolute lymphocyte counts and lymphadenopathy of the patients during venetoclax therapy. Day zero marks the start of the venetoclax treatment. Green lines show the time points of sample collection. Computer tomography (CT) scans for staging were performed at the time points marked by red lines. **b** Results from whole-exome sequencing are shown, including: number of somatic mutations, sample ploidy, percent of the genome undergoing copy number alterations (blue for losses and red for gains), and cancer-related gene mutations with pronounced clonal dynamics during therapy. Genomic alterations are annotated according to the color panel below the image. Sample type/compartment and the status if a patient has undergone a Richter's transformation are additionally indicated. Pre-treatment samples (T0) are shown in red. **c** Giemsa, Ki67, CD3, and CD274 stains from lymph node material of patient C811 after relapse from venetoclax. High protein levels of CD274 are consistent with the genomic amplification of the locus containing *CD274*. Scale bar, 100 μm

evolution (Fig. 2 and Supplementary Data). Patient C789 showed a linear evolution, where the clone harboring the *BTG1* (p.E46K) mutation was selected as the dominant clone in the sample at relapse (Fig. 2a). On the contrary, the other *BTG1*-mutated sample (C577) exhibited a branching evolution into three molecularly distinct lineages (Fig. 2b). The branch composed of subpopulation C1 and C2 was detected only in the pre-treatment sample (T0-peripheral blood [PB]). The sample harvested during treatment (T1-lymph node [LN]) only contained the branch of subclones C3 and C4. However, the branch carrying the *BTG1* (p. Q36H) mutation was only seen in the relapse sample (T2-LN). In addition, this branch also harbored a homozygous loss of *CDKN2A/B*. Taken together, the evolutionary trajectory of the clones carrying *BTG1* mutations (Fig. 2a, b) is in favor of an involvement of *BTG1* in the resistance to venetoclax treatment.

Patient C548 showed a divergent evolutionary path of two branches (Fig. 2c). One branch (subclone C3 and C4) was selected during venetoclax therapy. This branch harbored a homozygous loss of *CDKN2A/B*, and mutations in *BRAF* (p.K601E) and *MLL3* (p.S321fs), which first appeared in subclone C3 and were retained in its descendent C4. Therefore, all those alterations appear to be

clonal in the relapse sample (T1-PB). In contrast, the branch composed of subclone C1 and C2 was suppressed by venetoclax, even though it contained a frameshift deletion in the known CLL driver gene *BIRC3* (p.Q547fs).

Finally, case C586 showed a remarkable pattern of convergent evolution (Fig. 2d). We found two *SF3B1* mutations (c.1996A > C; p.K666Q and c.1997A > C; p.K666T) affecting the same codon, but evolved in two independent clones during venetoclax exposure (Supplementary Fig. 5). Both mutations were not detected in the pre-treatment sample by whole-exome sequencing. Digital droplet PCR, however, revealed that the *SF3B1* mutations were already present in extremely small subclones before treatment (T0-PB): a cancer cell fraction of 0.04% for p. K666Q and 0.033% for p.K666T. This shows that the convergent evolution of both *SF3B1* mutations occurred before the initiation of venetoclax therapy. Although both subclones carrying the *SF3B1* mutation were selected during treatment, they only reached a combined cancer cell fraction of 62% in the peripheral blood (T2-PB) and 96% in the bone marrow (T4-BM) sample at relapse. This suggests that the *SF3B1* mutations were not the single cause of venetoclax resistance in this patient. In contrast,

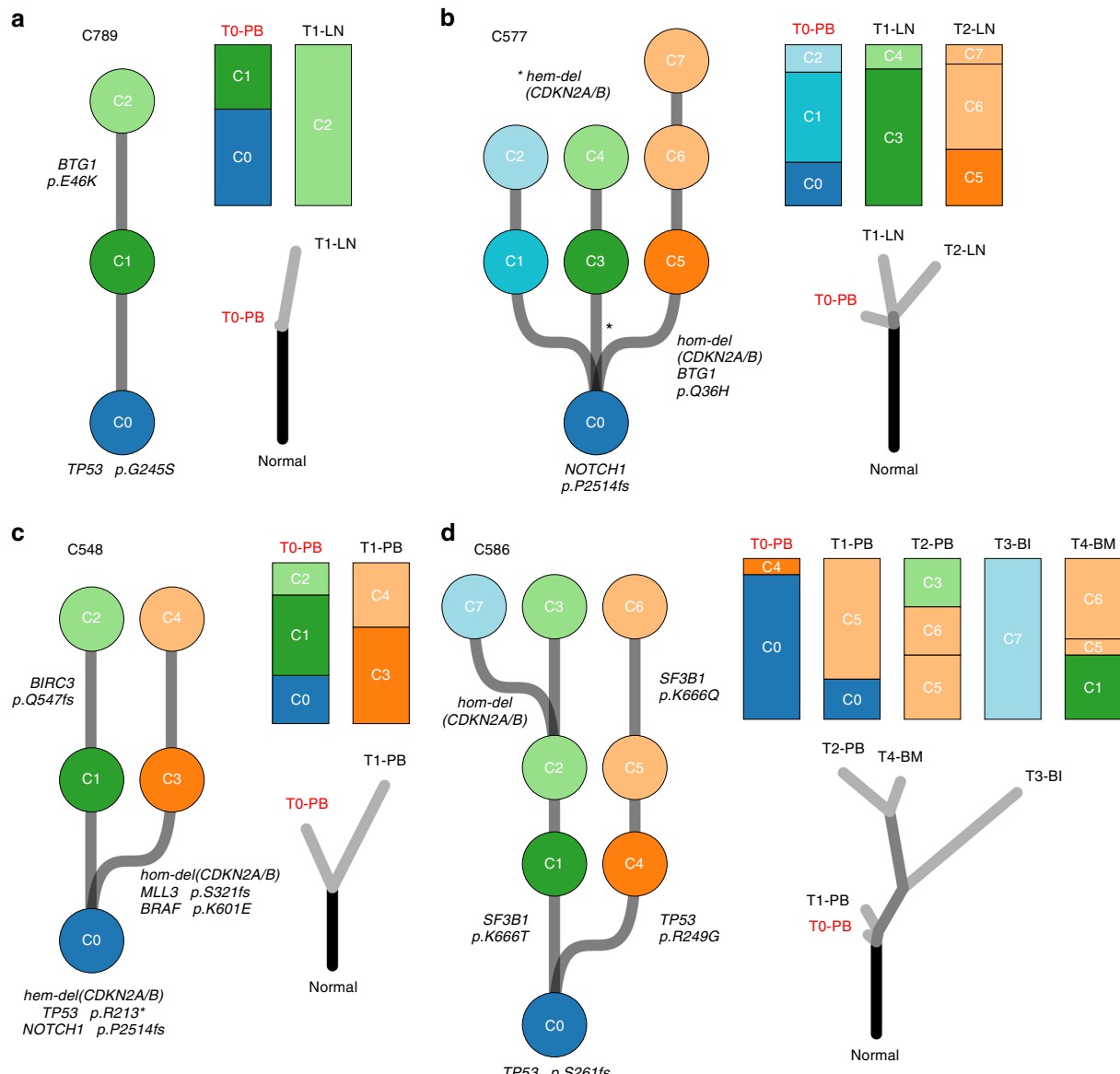

**Fig. 2** Heterogeneous clonal evolutions under venetoclax therapy. Phylogenetic trees at the left side of each panel demonstrate the clonal evolution of the reconstructed cell populations for each patient. Highlighted mutations that occurred during tumor evolution are present in all descendent clones. Therefore, mutations in the most common ancestor population (C0) are present in all analyzed samples at a clonal level. The second type trees (right-bottom of each panel) demonstrate the phylogenetic relations of the matched pre-treatment and relapse samples from a patient, as commonly used in other cancer evolution studies[10,13]. Clonal composition of the samples (top-right of each panel) provides a link between both types of phylogenetic trees. We inferred diverse evolutionary paths across the patients: **a** linear evolution (C789), **b** branching evolution into three lineages (C577), **c** divergent evolution of two branches (C548), and **d** convergent evolution (C586). Pre-treatment sample names are displayed in red. Notable gene alterations are shown in the context of the ancestral relation of the clones

the rapidly proliferating (Ki67-score of 90%) extranodal disease manifestation inside the right radial bone (T3-BI) was completely dominated by the single subpopulation C7, which carried a homozygous loss of *CDKN2A/B*.

**Functional analyses**. In order to assess if oncogenic BRAF signaling may induce venetoclax resistance, we overexpressed mutated *BRAF* (p.V600E) in a venetoclax-sensitive cell line OCI-LY19 (Fig. 3a). Exome sequencing of this cell line revealed a nonsense mutation in *CDKN2A/B* (p.W110*), as well as a genomic loss of one allele of *TP53* and a splice site mutation on the other allele. Therefore, this cell line presents a similar damage of key cancer-related genes, compared to patient C548. We found that the *BRAF*[V600E]-transduced cell line exhibited a pronounced

venetoclax resistance with a half-maximal growth inhibitory concentration (GI$_{50}$) larger than 10 μM in contrast to the empty vector control: GI$_{50}$ = 0.43 μM (Fig. 3b). In another venetoclax-sensitive cell line (U-2932) overexpression of the mutated *BRAF*[V600E] led only to a slight increase of the GI$_{50}$ value (from 0.72 μM for the empty vector control to 0.99 μM; Supplementary Fig. 6a). A reanalysis of published whole-exome-sequencing data[25] of U-2932 showed mutated *TP53* together with a loss of 17p, but no alterations in *CDKN2A/B*. In both cases, the expression of mutant *BRAF*[V600E] is paralleled by an increase in MCL1 protein levels. Next, we tested if a loss of *CDKN2A/B* alone can lead to venetoclax resistance and deleted the gene in the cell line OSU[26], using the CRISPR/Cas9 system. We found no difference in venetoclax sensitivity between OSU wild-type and

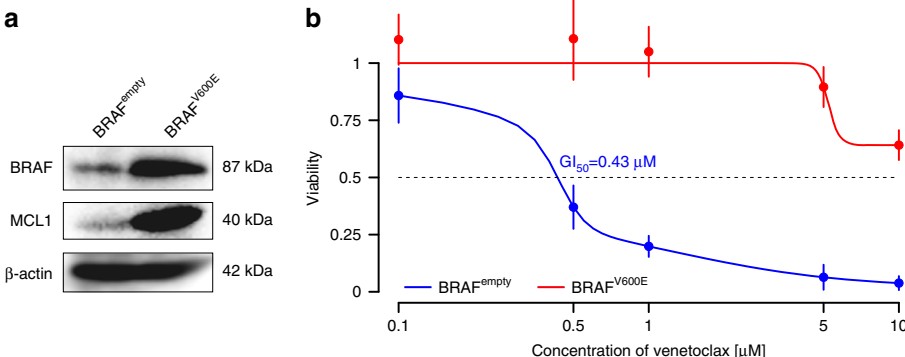

**Fig. 3** Overexpression of oncogenic *BRAF* in the OCI-LY19 cell line. **a** Western blot analysis of BRAF and MCL1 in the *BRAF*[V600E] overexpressing OCI-LY19 cell line vs. its empty vector control. **b** Growth inhibition of *BRAF*[V600E] transfected OCI-LY19 cells and the empty vector control is shown as a function of the concentration of venetoclax

*CDKN2A/B* knockout cells (Supplementary Fig. 6b). This result may explain why the *CDKN2A/B* homozygous deletions co-occured in all cases with other genomic alterations that developed during treatment.

## Discussion

CLL with dysfunctional p53, especially in the situation of relapse or refractory disease, has a particularly poor prognosis, and there are only few effective treatment options available for this high-risk category of patients. The BCL2 antagonist venetoclax has recently been approved for treatment of patients p53-deficient CLL (del(17p) or *TP53* mutations), who have failed or are not suitable for B-cell receptor (BCR) pathway inhibition with ibru-tinib or idelalisib. In addition, the drug can be used in patients, who have failed both chemo-immunotherapy and BCR pathway inhibition. Therefore, deciphering resistance mechanisms that arise during venetoclax therapy is of high clinical relevance.

We analyzed whole-exome-sequencing data of CLL specimens from eight patients before the initiation of venetoclax therapy and at the time of venetoclax resistance. All patients had shown a significant clinical response to venetoclax, before occurrence of disease progression or relapse. Our sequencing effort revealed intriguingly diverse patterns of clonal dynamics, such as linear, convergent, and divergent evolution under venetoclax treatment. In addition, all patients demonstrated signs of accumulating genomic instability, as demonstrated by an increasing number of acquired copy number alterations or aneuploidy. In most patient samples, we were able to detect alterations in cancer-related genes (i.e., *BRAF*, *CD274*, *NOTCH1*, *RB1*, *SF3B1*, and *TP53*) that evolved during venetoclax treatment. Surprisingly, genetic alterations in *BCL2* or functionally connected genes, such as *BAX* and *BAK* were not identified. Furthermore, we detected muta-tions in *BTG1* and homozygous deletions of *CDKN2A/B* as recurrent genomic events at the time of relapse under venetoclax exposure. *BTG1* might underlie an increased selection pressure under venetoclax therapy, as it seems to be required to maintain cell proliferation and its expression is regulated by BCL2[21,22]. Furthermore, our data suggests that complete loss of *CDKN2A/B* alone is not sufficient to induce venetoclax resistance.

Overexpression of oncogenic *BRAF* in lymphoma cell lines showed that it rendered lymphoma cells against venetoclax resistance, in vitro. Further functional analyses, however, point to a context-dependent interplay of genetically altered driver genes. Given the diverse genomic dynamics in our patient set investi-gated, it is likely that there are other/additional cellular mechanisms involved in multiple molecular patterns, which

ultimately lead to venetoclax resistance in CLL cells. Once large collectives of venetoclax-resistant CLL samples are available, further studies are required to elucidate these mechanisms. Notably, our sequencing effort was able to identify genome alterations at relapse (*BRAF*, *CD274*) that might qualify for fur-ther therapeutic options. Finally, comprehensive molecular test-ing is a suitable methodology to decipher mechanisms of venetoclax resistance and should be incorporated into future protocols to create molecularly-targeted salvage strategies for CLL patients under venetoclax treatment.

## Methods

**Patient sampling and nucleic acid extraction**. Between February 2014 and February 2016, we collected peripheral blood, bone marrow and tissue samples from eight patients, who presented with relapsed/refractory CLL upon oral vene-toclax therapy. The study was approved by the ethical review board of the Uni-versity of Cologne and performed according to the Declaration of Helsinki. All participants provided written informed consent.

Seven patients had received oral venetoclax as single agent therapy within the M13-982 2012-004027-20 trial[16] (NCT01889186). One patient (C789) had been treated with two cycles of bendamustine and obinutuzumab debulking (bendamustine 70 mg/m² day 1/2, q28d; obinutuzumab 100 mg/900 mg day 1/2, 1000 mg d8/15 in cycle 1, 1000 mg d1/8/15 in cycle 2–6, and q84d thereafter) with oral venetoclax added on day 1 of cycle 2 and maintained daily thereafter (CLL2-BAG study of the German CLL study group, EudraCT: 2014-000580-40, NCT02401503). After a weekly ramp-up schedule starting with 20 mg qod, all patients had achieved maximum dosing of venetoclax at 400 mg qod.

Baseline (T0) peripheral blood samples were obtained at a median of 12 days (range, 0–89) prior to the first dose of venetoclax. At this time all patients had CLL disease in peripheral blood detectable by flow cytometry according to the International Workshop on Chronic Lymphocytic Leukemia guidelines 2008[27], however, four patients presented a B lymphocyte count of <5.000/μl. Follow-up samples from peripheral blood, bone marrow or tissue were collected whenever the patient presented with signs of refractory or relapsing CLL disease and leftover material was available from diagnostic procedures. High purity (≥95%) CLL cells were separated from peripheral blood using a negative-selection of CD19-B-cells and column-based magnetic cell separation (Miltenyi, Bergisch-Gladbach, Germany). The CD19-negative non-B cells were collected as a matched non-cancer cell fraction ("normal") in each patient. At the time of refractory/progressive disease only in 3 of the 8 patients CLL-B cells could be sufficiently enriched from peripheral blood, as most patients did not present with a peripheral blood lymphocytosis at this time. In one case, a full lymph node biopsy (C577, T2-LN) was obtained at disease progression and CLL-B cells were selected via fluorescent labeling of CD5/19-positive cells (Biolegend, San Diego, CA, USA) and flow cytometry based cell sorting. In another case (C586, T2-BM) a bone marrow aspirate was available at relapse and CLL cells were enriched via positive selection of CD19-B cells (Miltenyi, Bergisch-Gladbach, Germany). In seven patients at least one punch biopsy of a lymph node or an extranodal disease lesion was obtained and used for nucleic acid extraction without additional cell separation due to the limited volume of tissue. Genomic DNA was purified from CLL-B-cell and non-B-cell fractions using standard columns (Qiagen, Hilden, Germany). All DNA specimens were confirmed to be of high molecular weight (>10 kb) by agarose gel electrophoresis.

**Whole-exome sequencing**. Whole-exome sequencing was performed from genomic DNA, using the SureSelect Human All Exon V6 (Agilent, Santa Clara, CA, USA) or the NimbleGen v2 target enrichment (Supplementary Table 2) according to the manufacturer's instructions. Obtained exome libraries were paired-end sequenced on either a HiSeq2000 ($2 \times 100$ bp) or a HiSeq4000 ($2 \times 75$ bp) platform (Illumina, San Diego, CA, USA). We achieved an average sequencing depth of 144X in the tumors and 90X in the normals.

**Analysis of whole-exome-sequencing data**. Raw sequencing reads were aligned to the human reference genome (NCBI build 37/hg19) using the BWA mem aligner (version 0.7.13-r1126). Concordant read pairs were masked as possible PCR duplicates and areas of overlapping read pairs were excluded from analysis in one read. We compared coverage differences between the two enrichment kits and filtered genomic partitions out that showed no sufficient coverage (<10X) in one of the kits to avoid kit-specific biases in the mutation calls. Somatic substitutions, insertions, deletions, copy numbers, and cellularity estimates were determined by our in-house cancer genome analysis pipeline[28–30].

**Reconstruction of clonal evolution**. Two different types of phylogenetic trees have been presented in this work. The first type of phylogenetic tree demonstrates the clonal evolution of reconstructed cell populations across cancer samples. The direction of evolution is shown from bottom to top and is rooted from the common ancestor clone. In each tree, nodes represent different clones that have existed during the evolution of the disease. Edges represent the ancestral relationships between clones. Mutations that distinguish the child clone from the mother clone are assigned to each edge. Here, phylogenetic tree reconstruction is performed by a novel algorithm, which is an extension of our previous approach[28]. The procedure consists of two steps: (1) inferring the mutation clusters that represent an evolutionary populations and (2) tree reconstruction based on these clusters. To infer mutation clusters based on cancer cell fractions of the mutations, we used a two-dimensional version of our mutation clustering method[28]. To reconstruct the tree, we first assume that the tumor samples are monoclonal, which means all cancer cells within a tumor are decedents of a single cell. We also assume that the mutations satisfy the infinite sites assumption, which states that a mutation occurs only once at a specific locus during evolution of the cancer. The flowing rules are considered to infer the tree from the cancer cell fractions of the mutation clusters: (1) connectivity rule (2) sum rule. The connectivity rule enforces that each node is connected to one mother node except the common ancestor (root) node. The sum rule ensures that the sum of cancer cell fractions over all child nodes is less than or equal to the cancer cell fraction of the mother node. Under these constraints our method automatically builds phylogenetic trees using linear integer programming.

The second type of trees are constructed based on maximum parsimony assumption using the Pars module form PHYLIP package[31] as it has been done by others[13]. Each tree shows the phylogenetic relations of the matched pre-treatment and relapse samples of a patient rooted from the normal cells of the patient. Leafs of the tree are different tumor samples from the same patient. Branches show the phylogenetic relations based on the somatic mutations difference of the samples. Length of each branch is proportional to the number of mutations assigned to the branch. The infinite sites assumption may not be fulfilled in the reconstruction of the second type trees.

**Methylation arrays**. DNA methylation was quantified using the HumanMethylationEPIC (EPIC) BeadChip (Illumina, CA, USA) according to the manufacturer's instruction. Raw IDAT files were processed in R (3.3.1) using the Bioconductor package *RnBeads* 1.6.1[32]. Methylation β-values were normalized using the *bmiq* method in order to correct for potential bias in DNA methylation measurements between Type I and Type II probes[33]. Principle component analysis was performed in R using the default values of the built-in function *prcomp*. Copy number profiles were derived by adapting our analysis tool for SNP 6.0 arrays[34] to methylation arrays.

**Digital droplet PCR**. Digital Droplet PCR was performed using the Bio-Rad QX200 ddPCR system (Bio-Rad, Hercules, CA, USA). The ddPCR probe mastermix and primers targeting distinct mutation sites or wild-type genes of interest were purchased from Bio-Rad. Obtained PCR raw data were processed using QuantaSoft v.1.6 (Bio-Rad).

**Production of retroviruses for *BRAF*$^{V600E}$ overexpression**. HEK293T cells (ATCC, Manassas, VA, USA) were plated into 10 cm dishes in Iscove's Modified Dulbecco's Medium (IMDM) + 20% fetal calf serum (FCS) + 50 mM beta-mercaptoethanol or RPMI + 20% FCS and incubated over night at 37 °C. Cells were co-transfected with pMDLg/pRRE und pMD2.G packaging plasmids and pBABEpuro expression plasmids encoding for BRAF V600E or empty vector using a standard calcium phosphate transfection protocol. The next day, medium was changed and cell culture supernatants were collected after 24, 48, and 72 h, centrifuged at $200 \times g$ for 5 min and sterile filtered. Using these viral supernatants human OCI-LY19 cells or human U-2932 cells (both kindly provided by Louis Staudt, National Cancer Institute, Bethesda, MD, USA) were transduced in the presence of 4 μg/ml polybrene (Santa Cruz Biotechnology, Dallas, TX, USA) for 24

h and selected with puromycin (Sigma-Aldrich, Munich, Germany). Finally, overexpression efficiency of BRAF and MCL1 expression was confirmed by immunoblotting. All cell lines used in these experiments were confirmed to be mycoplasma negative and authenticated by short-tandem repeat profiling.

**Cell viability measurement**. OCI-LY19 or U-2932 cells were plated into sterile 96-well plates at 10,000 cells per well. Twenty-four hours after seeding, cells were treated with various concentrations of venetoclax (Selleckchem, Munich, Germany) for 72 h. After completion of drug exposure, relative cell viability was determined by measuring the ATP content in each well (CellTiter-Glo®; Promega, Madison, WI, USA) and is normalized to a control treated with the vehicle solution.

**Cell culture: CDKN2A CRISPR screen**. HEK293 cells (DSMZ, Braunschweig, confirmed mycoplasma negative, not further authenticated) were cultured in Dulbecco's modified Eagle's medium (DMEM; Gibco, Thermo Fisher Scientific, Waltham, MA, USA) supplemented with 10% FCS (Gibco) 1% penicillin–streptomycin (v/v) (Gibco). CLL derived OSU cells (kindly provided by John Byrd, Ohio State University Comprehensive Cancer Center, Columbus, OH, USA, confirmed mycoplasma negative, not further authenticated) were cultured in RPMI 1640 (Gibco) supplemented with 10% FCS and 1% penicillin-–streptomycin (v/v) (Gibco). Cells were incubated at 37 °C in a humidified 5% $CO_2$ atmosphere.

**DNA constructs**. Eight 20-nt DNA sequences (see below) in exons 1 and 2 in the human genomic *CDKN2A* locus were selected for producing single-guide RNA for CRISPR-associated DNA endonuclease targets by using a publically available resource (http://crispr.mit.edu). The lentiCRISPR v2 vector[35] was purchased from Addgene (Cat. #52961). The oligos were annealed with bottom oligos and cloned into lentiCRISPR v2 restricted by *BsmB1* (#R0580S; New England Biolabs, Boston, MA, USA). All generated constructs were analyzed by DNA sequencing using a primer specific to the U6 promoter.

| Sequence | Target site |
|---|---|
| 5′-CACCGACCGTAACTATTCGGTGCGT-3′ | Exon 1 |
| 5′-CACCGGGCCTCCGACCGTAACTATT-3′ | Exon 1 |
| 5′-CACCGCACCGAATAGTTACGGTCGG-3′ | Exon 1 |
| 5′-CACCGAGCACCGAATAGTTACGGT-3′ | Exon 1 |
| 5′-CACCGGGTACCGTGCGACATCGCGA-3′ | Exon 2 |
| 5′-CACCGACCTTCCGCGGCATCTATGC-3′ | Exon 2 |
| 5′-CACCGTGGGCCATCGCGATGTCGCA-3′ | Exon 2 |
| 5′-CACCGGCCCGCATAGATGCCGCGGA-3′ | Exon 2 |

**Lentivirus production and transduction**. The lenticCRISPR v2 inserted with single-guide RNA, together with psPAX2 packaging plasmid (Addgene, Cambridge, MA, USA Cat. #12260) and pMD2.G envelope plasmid (Addgene Cat. #12259) DNA were combined together and transfected into HEK293 cells by using the Lipofectamine 2000 reagent (Cat. #11668-019, Invitrogen, Thermo Fisher Scientific, Waltham, MA, USA). The transfection solution was added dropwise to the cells and incubated for 6 h at 37 °C in a humidified 5% $CO_2$ cell culture incubator. Six hours post transfection the medium was replaced by adding DMEM supplemented with 5% FCS and 1% penicillin–streptomycin and incubated for 24 h. Eighteen hours post medium replacement, sodium butyrate (Cat. #B5887; Sigma-Aldrich) was added to the culture medium at a final concentration of 1 mM. The viral harvest was performed for two times in 24-h intervals. After each collection, the virus-containing medium was cleared by centrifugation at $300 \times g$ for 5 min at 4 °C, filtered through a 0.45 μm filter unit (Cat. # 10462100; Whatman, GE Healthcare Life Sciences, Buckinghamshire, UK) and stored at −80 °C until use. Lentivirus-containing medium was used to infect OSU cells. For this purpose, $10^7$ cells were resuspended in viral supernatant supplemented with 2 μg/ml polybrene and centrifuged at $800 \times g$ for 2 h and 32 °C. After spinoculation the cell pellet was resuspended and cells were incubated for further 24 h under viral containing conditions at 37 °C in a humidified 5% $CO_2$ atmosphere. After incubation, cells were washed with PBS, centrifuged and resuspended in cell culture growing medium. Cells were cultivated for 4 to 5 days until start of selection by 0.5 μg/ml puromycin (Cat. #ant-pr-1; InvivoGen, San Diego, CA, USA) for 72 h. Subsequently 85 cells were resuspended in 20 ml of medium and distributed by pipetting 200 μl of this cell suspension into 96 wells. After outgrowth of single-cell clones, these clones were lysed and prepared for western blotting.

**Flow cytometry**. Apoptosis was determined by flow cytometry using AnnexinV-FITC (Immuno Tools, Friesoythe, Germany)/7AAD (eBioscience, San Diego, CA, USA) staining in AnnexinV staining buffer (BD, Heidelberg, Germany). Measurement was carried out by fluorescence-activated cell sorting (FACS) Canto flow cytometer (BD). Double negatives were considered as viable. Data was analyzed by FACS DIVA Software (BD).

**Western blot and immunodetection**. After harvesting, cells were washed in ice-cold PBS, subsequently cells were lysed with RIPA III buffer (50 mM TRIS-HCl pH 8, 150 mM NaCl, 0.1% sodium dodecyl sulfate (SDS), 0,5% DOC, 1% NP-40) supplemented with protease inhibitor (#P8340, Sigma-Aldrich) and phosphatase inhibitor cocktail (#P5726; #P0044, Sigma-Aldrich). Protein content was determined by Roti-Quant (#K.015.1, Roth, Karlsruhe, Germany). Cells were mixed with SDS-loading buffer (60 m M Tris-HCl pH 6.8, 3.3% SDS, 20 mM Dithiothreitol (DTT), 0.01% bromphenol blue), then mixture was heated to 95 °C for 5 min. In the following protein samples were loaded on a NuPAGE 5–12% Bis-Tris gel (Thermo Fisher Scientific), separated by gel electrophoresis at 80 V for 3 h and then transferred to a Protran nitrocellulose membrane (0.45 μM pore size, Whatman, Maidstone, UK) by tank-blotting at 80 V for 1 h. Afterwards the blotting membrane was blocked with 5% non-fat dry milk in tris-buffered saline for 1 h. Blots were probed with the following antibodies against: CDKN2A (#ab81278, Abcam, Cambridge, UK, 1:1000), BRAF (sc-5285, Santa Cruz, 1:500), and MCL1 (CST-5453, Cell Signaling, Danvers, MA, USA, 1:1000). β-actin (#A5316, Sigma-Aldrich, 1:10,000) was used as loading control. Detection of primary antibodies was achieved via specific IRDye secondary antibodies (LI-COR Biosciences, Lincoln, NE, USA, 1:20,000–1:10,000). Protein bands were detected by LI-COR reader (LI-COR) and quantified by LI-COR Software (LI-COR). Full-length images of the most important western blots (Fig. 3 and Supplementary Fig. 6b) are shown in Supplementary Fig. 7.

**Data availability**. Sharing of the exome-sequencing data of this study outside our institution is not permitted due to restrictions in our informed patient consent. Compatible with our patient consent is to release a data set that contains information of somatic mutations. Therefore, we provide tables of all somatic mutation calls and our entire copy number analysis in the Supplementary Data. Mini bam-files around the positions of somatic mutations are uploaded to the European Nucleotide Archive (https://www.ebi.ac.uk/ena) under accession number: PRJEB24344. These data sets are sufficient to reproduce our findings and offer the possibility to apply, e.g., alternative phylogenetic reconstruction methods. Mutation data of untreated CLL patients were downloaded from Landau et al.[3], Nature, 2015 (doi: 10.1038/nature15395) and Puente et al.[8], Nature, 2015 (doi: 10.1038/nature14666).

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

## Acknowledgements

We are indebted to our patients for making available the specimens that were analyzed within this study. We also like to thank William Pao for fruitful discussions and Laura Wilden for her contribution to collect and prepare samples for sequencing. This work was supported by the Deutsche Forschungsgemeinschaft (DFG, CRU-286) to C.D.H., K.-A.K., H.C.R., M.H., L.P.F., M.P., and (DFG, JA 2439/1-1) to R.D.J., the German Ministry of Science and Education (BMBF) as part of the e:Med initiative (grant no. 01ZX1303A to H.C.R., U.L., and M.P. and grant no. 01ZX1406 to M.P.), the Volkswagenstiftung (Lichtenberg Program, H.C.R.), the Else Kröner-Fresenius Stiftung (EKFS-2014-A06, H. C.R.), the Deutsche Krebshilfe (111724, H.C.R.), German José Carreras Leukemia Foundation (grant no. DJCLS R 13/33, K.-A.K. and DJCLS R12/26, L.P.F and H.C.R.), Roche research funding (L.P.F. and C.D.H.), the Stiftung Kölner Krebsforschung (L.P.F. and H.C.R.), the Gusyk family support program at the University of Cologne (C.D.H.), and the Center for Molecular Medicine Cologne (CMMC).

## Author contributions

Conception and design: C.D.H., N.A., H.C.R., M.H., L.P.F., M.P. Provision of study materials and patients: C.D.H., P.M., N.K., M.S., P.C., C.-M.W., T.P., A.S., C.P., B.E., R. B., K.-A.K. Conduct of the experiments: C.D.H., J.W., A.S., R.D.J., O.M., S.O., P.M., V.B., D.T., J.A., P.N., R.C., S.C.S. Data analysis, and interpretation: C.D.H., N.A., M.C., V.A., U.L., H.C.R., L.P.F., M.P. Manuscript writing: C.D.H., N.A., L.P.F., M.P. All authors read and approved the final manuscript.

## Additional information

**Competing interests:** C.D.H, C.-M.W., B.E., K.-A.K., M.H., and L.P.F. received research funding from Hofmann-La Roche. Research support was also provided by AbbVie to P.C., C.-M.W., B.E., K.-A.K., and M.H. P.C., C.-M.W., B.E., K.-A.K., H.C.R., M.H., and L.P.F. obtained consulting and/or speaker's honoraria from AbbVie. P.C., C.-M.W., B.E., K.-A.K., and M.H. received consulting and/or speaker's honoraria from Hofmann-La Roche. AbbVie provided travel support to P.C., N.K., and L.P.F. The remaining authors declare no competing financial interests.

