## [Peer Review File · Nature Communications]

Reviewer #1 (Remarks to the Author):

The paper by CD Herling and coworkers investigates the molecular basis of the clonal dynamics leading to the development of venetoclax resistance in chronic lymphocytic leukemia (CLL). As such this study falls within the critical field of understanding the mechanisms that cause treatment resistance in CLL and more generally in cancer.

By WES the Authors identified recurrent mutations in BTG1 and homozygous deletions affecting CDKN2A/B that developed during venetoclax treatment, as well as a mutation in BRAF and a high-level focal amplification of CD274 (PD-L1) that might pinpoint molecular aberrations and as such potentially offer suggestions for further therapeutic interventions.

The paper is well written and well thought out, the experimental plan is impeccable and the data are of great interest for all those who are interested in increasing their knowledge in the fundamental biology of CLL. That said, and considering that the Authors apparently underline the clinical relevance of their data, the obvious question becomes: which is the real clinical implication of these findings? In other words: what are the Authors suggesting to do to clinicians involved in the treatment of CLL patients? Would some patients have to be spared venetoclax treatment? I guess no, but, if so, how can we identify them? Are the Authors suggesting that their findings lead to the possibility of alternative treatments for patients relapsing after venetoclax treatment? I believe the answer is yes, however and again the question becomes: how can we identify these patients without undergoing the painstaking procedure of WES which is not available to every Center?

I would consider important to have an answer to these questions and especially to have the Authors spell out a clear position in the paper

Reviewer #2 (Remarks to the Author):

The present manuscript reports exome sequencing analysis 8 CLL patients on samples obtained before and after treatment and reports patterns of clonal evolution whilst on therapy as well as recurrent deletions in CDKN2A/B (3 patients) as well as mutations in BTG1 (2 patients). The authors postulate that these events developed during treatment and may be associated with acquired resistance to venetoclax and propose that these may identify targets for therapeutic intervention.

The manuscript is straightforward and well written. The observation is intriguing but limited in numbers and power and the authors do not go into much effort to relate their findings to the larger sequencing studies in CLL. For example how often do we see these secondary events in primary / diagnostic CLL samples not undergoing Venetoclax treatment?

The authors postulate that their findings highlight novel therapies. This is in relation to the BRAF mutation and CD274. The authors do not mention or comment that these mutations are confined in branches or subclonal compartments. In relation to PD-L1, we now know so much more on predictors of response to checkpoint blockade inhibitors such as neo-antigen load and burden, that the argument made in the manuscript is a bit weak.

Figure 1b shows patient C586 has three hits on TP53 when TP53 deletions are also considered. There is also a discrepancy in the calls for the splice site TP53 mutation between the samples. Discrepancies in calls of samples taken at secondary timepoints from the same individual are seen elsewhere in the figure. Could the authors comment on what happened?

Patient 3 – I would edit the sentence Data from methylation arrays – to copy number analysis using methylation array derived data validated...

Reviewer #3 (Remarks to the Author):

This manuscript evaluates the tumor genomic changes pre, during, and post venetoclax therapy in patients with chronic lymphocytic leukemia (CLL) to determine genetic causes underlying resistance to venetoclax. The authors performed paired WES and methylation arrays in 8 CLL patients at two or more time points.

Although the concept of the study is of scientific interest, this manuscript is descriptive, sample size is too small, and the results do not justify the authors' claim that the identified alterations strongly suggest an important role in acquired venetoclax resistance. Further studies and larger sample sizes are needed.

Other comments:

The authors evaluated somatic mutations in key cancer-related genes and identified non-synonymous mutations (Figure 1b) that they validated using other technology (Supplementary Figure 3); however, not all of the mutations identified are shown in Supplementary figure 3, including the mutations highlighted in the text (e.g., CD274)

Supplementary Figure 2: The authors should justify how they defined that the methylation results within patients were clustering in close proximity. Given the range of scale and a number of subjects whose sample results are spread out, I disagree with the authors assessment

Two capture kits were used to perform WES (Agilent and NimbleGen). Given that capture kits do not have 100% overlap of regions, a mutation not observed on one capture kit could be due to that capture kit not capturing that mutation or it was poorly captured. Thus more detail is needed as to what was done. E.g., For each sample, was the same capture kit used? Or were the pre treatment capture kits all done with one capture kit and all the post treatment done with the other capture kit.

Point-by-point response to the reviewer comments:

Reviewer #1:

The paper by CD Herling and coworkers investigates the molecular basis of the clonal dynamics leading to the development of venetoclax resistance in chronic lymphocytic leukemia (CLL). As such this study falls within the critical field of understanding the mechanisms that cause treatment resistance in CLL and more generally in cancer.

By WES the Authors identified recurrent mutations in BTG1 and homozygous deletions affecting CDKN2A/B that developed during venetoclax treatment, as well as a mutation in BRAF and a high-level focal amplification of CD274 (PD-L1) that might pinpoint molecular aberrations and as such potentially offer suggestions for further therapeutic interventions.

The paper is well written and well thought out, the experimental plan is impeccable and the data are of great interest for all those who are interested in increasing their knowledge in the fundamental biology of CLL. That said, and considering that the Authors apparently underline the clinical relevance of their data, the obvious question becomes: which is the real clinical implication of these findings? In other words: what are the Authors suggesting to do to clinicians involved in the treatment of CLL patients? Would some patients have to be spared venetoclax treatment? I guess no, but, if so, how can we identify them? Are the Authors suggesting that their findings lead to the possibility of alternative treatments for patients relapsing after venetoclax treatment? I believe the answer is yes, however and again the question becomes: how can we identify these patients without undergoing the painstaking procedure of WES, which is not available to every Center? I would consider important to have an answer to these questions and especially to have the Authors spell out a clear position in the paper.

We would like to thank the reviewer for the favorable evaluation of our manuscript and we fully agree with the reviewer's point that the actual clinical implications of our findings have to be discussed more deeply in the manuscript. We have therefore revised the discussion on page 6 and 7 of our manuscript to include the following aspects:

The patient set we have investigated (patients with mutated and/or deleted *TP53*, relapsed/refractory disease) resembles a clinically particularly challenging patient population, which usually presents refractory to chemo- or immunotherapy based treatment options and has a highly unfavorable prognosis. The BCL2-antagonist venetoclax has demonstrated substantial activity in this high-risk patient population with approximately 80% responders, including ~10% of complete remissions (Stilgenbauer S et al. *Lancet Oncol* 2016; Roberts AW et al. *N Engl J Med* 2016). This level of efficacy is superior to conventional chemo- or chemo-immunotherapy and is expected to improve survival of these patients. Thus, venetoclax represents an important treatment option in the otherwise very limited drug armamentarium for such high-risk CLL patients. However, long-term follow-up data still have to be awaited to actually determine the final clinical role and most efficient usage of this drug in patients. Furthermore, systematic studies on molecular mechanisms of venetoclax treatment resistance in primary CLL specimens are not available to date.

All patients included in our study demonstrated a significant clinical benefit from venetoclax therapy before progression/relapse occurred. Four of the 8 patients (50%) presented with Richter's transformation (RT) at relapse. This rate appears to be expectable according to reports from recent phase I and phase II trials (43,9% of

RT in Roberts AW et al. *N Engl J Med* 2016; 45,8% of RT in Stilgenbauer S et al. *Lancet Oncol* 2016), which investigated venetoclax monotherapy in relapsed/refractory CLL, including patients with *TP53* aberrations. Patients with genetic *TP53* lesions are prone to develop Richter's transformation (Fabbri G et al. *J Exp Med* 2013; Chigrinova E et al. *Blood* 2013). Thus, our study includes a clinically representative set of patients with and without transformed disease at the time of relapse after venetoclax.

We did not identify a recurrent mutation in the drug target itself or in a molecule in close molecular or pathway-related proximity to *BCL2*. This is in contrast to cell line data, where mutations in *BCL2* itself or *BAX* were detected under venetoclax treatment pressure (Fresquet V et al. *Blood* 2014, Tahir SK et al. *BMC Cancer* 2017). The fear of target mutations has been sparked by the experiences obtained with BCR-ABL-kinase inhibitors in CML, and also the recent identification of mutations in the drug target BTK in patients under ibrutinib therapy (Woyach JA et al. *N Engl J Med* 2014). Although the size of our study prohibits exclusion of such target mutations to occur under targeted *BCL2*-inhibition, it seems to be unlikely, that such alterations are the predominant mechanism of venetoclax resistance in *TP53*-deficient CLL. Our study was also not designed or powered for the identification of a genetic marker of resistance, which would allow predictive statements upon its detection for treatment outcome.

Instead, our study gives exemplary, but important insights into the highly diverse evolutionary dynamics and individual trajectories of CLL cells under venetoclax-mediated selection pressure *in vivo*. The accumulation of ongoing DNA damage plus the selective evolution of subclones, which are able to recruit one or more other pro-tumorigenic driver lesions (i.e., *BRAF*, *CDKN2A/B*, *CD274*, *NOTCH1*, *SF3B1*, *TP53*), seem to contribute to a spectrum of multiple molecular patterns ultimately leading to venetoclax resistance.

Overall, neither from a clinical nor from a molecular standpoint it can be recommended from our data to spare any patients from venetoclax therapy. The development of transformed disease seems to be a common complication in *TP53*-deficient CLL. Whether the rate of RT under venetoclax is higher than under other treatments has to be clarified within prospective clinical trials, including higher numbers of patients.

Importantly, our analysis demonstrates that DNA sequencing may be beneficial to identify evolving genetic lesions that might qualify the patient for further treatment options. Thus, we propose that comprehensive molecular testing, at least of potentially druggable molecular targets, should be considered in further clinical trials investigating the use and outcome of venetoclax-based therapies. As current molecular technologies and their sensitivities are rapidly evolving, we preferred to not recommend a certain technology. However, for clinical practice this could mean that in the future we would monitor CLL cases with high-risk (i.e. *TP53* dysfunction) by molecular tests, including targeted sequencing for actionable lesions, such as activating *BRAF* mutations or *CD274* amplifications. This approach would use peripheral blood, since most CLL cases have circulating leukemia cells available, and in the rare cases of patients without lymphocytosis, liquid biopsy technologies could be used to monitor circulating DNA (Yeh P et al. *Nat Commun* 2017). Required technologies should be investigated systematically within prospective clinical trials.

Reviewer #2:

The present manuscript reports exome sequencing analysis 8 CLL patients on samples obtained before and after treatment and reports patterns of clonal evolution whilst on therapy as well as recurrent deletions in *CDKN2A/B* (3 patients) as well as mutations in *BTG1* (2 patients). The authors postulate that these events developed during treatment and may be associated with acquired

resistance to venetoclax and propose that these may identify targets for therapeutic intervention.

The manuscript is straightforward and well written. The observation is intriguing but limited in numbers and power and the authors do not go into much effort to relate their findings to the larger sequencing studies in CLL. For example how often do we see these secondary events in primary / diagnostic CLL samples not undergoing Venetoclax treatment?

We thank the reviewer for this important comment/suggestion. Therefore we analyzed data from 559 treatment-naïve CLL samples, downloaded from two published large-scale sequencing projects (Landau et al. *Nature* 2015 and Puente et al. *Nature* 2015). Please note, that we removed the samples from the Spanish ICGC CLL project, which were already included in the dataset of Landau et al. to avoid possible patient duplicates with the data provided by Puente et al. This analysis revealed that non-synonymous mutations affecting *BTG1* rarely occurred in this dataset (only in 3 of the 559). We therefore conclude that a spontaneous development of *BTG1* mutations without a positive treatment selection is highly unlikely with a probability of 7.8×10^{-4} . Furthermore, we could not find a sample with a homozygous deletion of *CDKN2A/B* in the dataset of Puente et al. (n=125), where copy number information was available. However, recent studies have found an association between homozygous deletions of *CDKN2A/B* and Richter's transformation (Chigrinova et al. *Blood* 2013, Fabbri et al. *J Exp Med* 2013). In line with these findings, two of the three samples that developed homozygous deletions of *CDKN2A/B* have shown a Richter's transformation at the time of venetoclax relapse. Furthermore, a CRISPR/Cas9 knockout of *CDKN2A* in the OSU cell line showed no effect on venetoclax sensitivity (**Supplementary Figure 6b**), which might explain why the observed homozygous deletions of *CDKN2A/B* in our patient collective always co-evolved with other driver genes. *BRAF* mutations, on the other side, are relatively frequent in untreated CLL samples (21 of the 559 samples). The high frequency and the mutation pattern are compatible with the notion that *BRAF* is a potential driver gene in CLL if mutated (Landau et al. 2015). We added these findings in the results section (page 4) of the revised manuscript.

The authors postulate that their findings highlight novel therapies. This is in relation to the BRAF mutation and CD274. The authors do not mention or comment that these mutations are confined in branches or subclonal compartments. In relation to PD-L1, we now know so much more on predictors of response to checkpoint blockade inhibitors such as neo-antigen load and burden that the argument made in the manuscript is a bit weak.

This is an important point since it shows that our description of the phylogenetic tree of the subclones was not sufficient and can lead to misunderstandings. In particular, the labeling of the mutations may lead to confusions because they are not only present in the subclone they first appeared but are also present in all of its descendants. Mutations present in the root node: C0 are therefore in all subclones and samples. These mutations thus constitute the most common ancestor. Hence, the *BRAF* mutation is present in subclone C3 and C4. At relapse, the branch composed of C3 and C4 is completely selected, such that the *BRAF* mutation is clonal at relapse. We added more description of the phylogenetic tree in the revised manuscript (to page 5 and to the legend of **Figure 2**).

In order to strengthen the result that an oncogenic variant of *BRAF* may be involved in the venetoclax relapse, we overexpressed *BRAF*^{V600E} in two cell lines that are initially susceptible to venetoclax. We found that the half-minimal growth inhibitory concentration is substantially increased in one *BRAF*^{V600E}-transduced cell

line (OCI-LY19), whereas the second cell line only showed a small increase of half-minimal growth inhibitory concentration (U-2932; **Figure 3** and **Supplementary Figure 6a**). For both cell lines, the overexpression of the *BRAF*^{V600E} variant is paralleled by an increase of MCL1 protein expression.

In our case, the clonality of the *CD274* amplification cannot be inferred from the sequencing data, because the high-level amplification of *CD274* leads to an unfavorable signal-to-noise ratio, which prohibits reliable calls of subclonal copy number changes in this region. However, amplifications of *CD274* have been observed in several other cancer types other than CLL (mostly solid tumors). This suggests a certain selective pressure causing CLL cells of patient C811 to develop the amplification of *CD274*. At the present time, we do not have clear evidence to predict a response to immune checkpoint inhibitors in this patient. Unfortunately, a clinical verification of an immune therapy is not possible because the patient deceased in the meantime, due to a septic complication after allogeneic stem cell transplantation. To account for this uncertainty, we rephrased our statement from: "... suggests the use of immune checkpoint inhibitors..." to "... may be susceptible to immune checkpoint blockade..." in the revised manuscript.

Figure 1b shows patient C586 has three hits on TP53 when TP53 deletions are also considered. There is also a discrepancy in the calls for the splice site TP53 mutation between the samples. Discrepancies in calls of samples taken at secondary time points from the same individual are seen elsewhere in the figure. Could the authors comment on what happened?

Similarly to the previous comment, the discrepancies are due to our misleading mutation labels. The frame-shift deletion in *TP53* (p.S261fs) is present in all samples of patient C586 since it is contained in the most common ancestor node. The p.R249G mutation on the other side is subclonal and present in the complete branch composed of C4, C5, and C6. Together with the deletion of the other allele, *TP53* is hit in a bi-allelic fashion in all samples.

Patient 3 – I would edit the sentence Data from methylation arrays – to copy number analysis using methylation array derived data validated...

In the revised manuscript, we changed the sentence: "...Methylation arrays were additionally used to validate these copy number changes..." to: "...Copy numbers from exome sequencing were validated by the methylation arrays..."

Reviewer #3:

This manuscript evaluates the tumor genomic changes pre, during, and post venetoclax therapy in patients with chronic lymphocytic leukemia (CLL) to determine genetic causes underlying resistance to venetoclax. The authors performed paired WES and methylation arrays in 8 CLL patients at two or more time points.

Although the concept of the study is of scientific interest, this manuscript is descriptive, sample size is too small, and the results do not justify the authors' claim that the identified alterations strongly suggest an important role in acquired venetoclax resistance. Further studies and larger sample sizes are needed.

We agree with the reviewer that the sample size is quite small. However, it is unrealistic to obtain further samples from our center/network in the immediate future, as venetoclax is now typically used as part of combination regimens in our

center/study group. Moreover, in order to compensate the rather small sample size, we analyzed sequencing data from published treatment-naive samples (as also suggested by reviewer 2). Here, we found that non-synonymous *BTG1* mutations are extremely rare (only 3 in 559 samples). Using this rate, we estimated that the likelihood that the *BTG1* spontaneously evolved without a selective pressure by the therapy is 7.8×10^{-4} in our data set. Similarly, we could not find homozygous deletions of *CDKN2A/B* in sequenced pre-treatment samples where copy number data was available (n=125). Recently, homozygous deletions of *CDKN2A/B* have been associated with Richter's transformation (Chigrinova et al. *Blood* 2013, Fabbri et al. *J Exp Med* 2013). In line with this observation, 2 of the 3 patients that developed homozygous deletions of *CDKN2A/B* during venetoclax therapy also showed a Richter's transformation at the time of relapse.

In addition to this meta-analysis of published data we carried out a series of functional experiments to gain further evidence if some of the observed alterations can confer venetoclax resistance *in vitro*. First, we overexpressed the oncogenic *BRAF* variant p.V600E in the lymphoma cell lines OCI-LY19 and U-2932. While overexpression of *BRAF*^{V600E} was massively inducing venetoclax resistance in the OCI-LY19 cell line, the effect was much smaller for U-2932 (**Figure 3** and **Supplementary Figure 6a**). Genomic profiling of these cell lines revealed that both cell lines harbored *TP53* mutations but only OCI-LY19 showed a damaging mutation in *CDKN2A/B* (p.W110* nonsense mutation) and presents therefore a similar damage in key cancer-related genes as patient C548. For both cell lines, overexpression of the *BRAF*^{V600E} variant is paralleled by an increase of MCL1 protein expression. Next, we investigated if loss of *CDKN2A* alone can induce venetoclax resistance. To this end, we performed a CRISPR/Cas9 knockout of *CDKN2A* in the OSU cell line. We found that knocking out *CDKN2A* in OSU cells did not alter the sensitivity of venetoclax (**Supplementary Figure 6b**). This might explain why the observed homozygous deletions of *CDKN2A/B* in our patient collective always co-evolved with other driver genes.

Furthermore, our study also demonstrates that DNA sequencing may be beneficial to identify evolving genetic lesions (*BRAF*, *CD274*) that might qualify the patient for further treatment options. Given the large intra-tumor heterogeneity of CLL and the limited treatment options available for patients with *TP53* lesions, we believe, that these findings are of high clinical and scientific interest. Although these druggable alterations are present only in single patients, they demonstrate the potential use of genetic profiling to tailor salvage treatment options in a subset of CLL patients with very poor prognosis.

Other comments:

The authors evaluated somatic mutations in key cancer-related genes and identified non-synonymous mutations (Figure 1b) that they validated using other technology (Supplementary Figure 3); however, not all of the mutations identified are shown in Supplementary figure 3, including the mutations highlighted in the text (e.g., CD274).

Thanks for pointing out that we have only shown a subset of our mutation validations of the key cancer-related genes. We have now added our validation results from the digital droplet PCR of the *NOTCH1* frameshift mutations to **Supplementary Figure 3** and we validated (using conventional dideoxy sequencing) all *TP53* mutations that were not detected as part of the previously published study: Stilgenbauer S et al. *Lancet Oncol* 2016. In addition, we also validated the *MLL3* mutation occurring in patient C548 with dideoxy sequencing. Due to a lack of further genomic material we were not able to confirm the *BIRC3* mutation. All validations of these point mutations are now shown in **Supplementary Figure 3** of the revised manuscript. Validations of

copy number changes, such as homozygous deletions of *CDKN2A/B* and the amplification of *CD274* are presented in **Supplementary Figure 4**.

Supplementary Figure 2: The authors should justify how they defined that the methylation results within patients were clustering in close proximity. Given the range of scale and a number of subjects whose sample results are spread out, I disagree with the author's assessment.

We thank the reviewer for pointing out this important issue and agree that our assessment is inconclusive. This is mainly due to the predominant sampling of lymph node material in the relapse situation. In contrast, all samples prior venetoclax therapy was derived from peripheral blood. Since lymph node specimens are less pure than the FACS sorted blood samples. The difference in purity hampers a robust assessment of methylation changes by the venetoclax treatment. For only two patients (C626 and C586) we had relapse samples that were derived from the peripheral blood. Supplementary Figure 2 shows that these samples cluster together, but in order not to overstate this finding from a small subset, we changed our assessment of the results from the methylation analysis in the revised manuscript to: *"...At time of relapse, lymph node specimens with a lower purity were mostly available in contrast to the generally pure pre-treatment samples derived from peripheral blood (Fig. 1b, Supplementary Table 2). This hampers a robust assessment of treatment-specific changes in the methylation profiles, since most of the variability seen in the methylation patterns within each patient might be due to differences in the compartments analyzed (Supplementary Fig. 2)..."*.

Two capture kits were used to perform WES (Agilent and NimbleGen). Given that capture kits do not have 100% overlap of regions, a mutation not observed on one capture kit could be due to that capture kit not capturing that mutation or it was poorly captured. Thus more detail is needed as to what was done. E.g., For each sample, was the same capture kit used? Or were the pre treatment capture kits all done with one capture kit and all the post treatment done with the other capture kit.

This is an important point. We have therefore added the information of which exon enrichment kit was used in the revised manuscript. This shows that for only one patient (C586) different enrichment kits were used. In addition, we have carefully checked if there was enough coverage in all samples for all mutations that were discussed in the manuscript. Therefore, we can exclude the possibility that our results were biased by differences in the enrichment kits used or by differences in the local coverage distribution.

Reviewer #1 (Remarks to the Author):

C.D. Herling and co-workers have satisfactorily to all my queries

Reviewer #2 (Remarks to the Author):

In their manuscript revision the authors have presented a thoroughly curated dataset and analysis of their findings but the implication of these is very limited in scope.

A few minor points.

Their comparison with external and publicly available datasets do suggest that there is an enrichment of BTG1 mutations.

Their clonal representation on Figure 2 should provide information on all mutations that delineate new clones. In its present form not all nodes are defined by distinct mutations. The authors should consider revising.

Supplementary Figure 2

The authors postulate that methylation differences are mostly between different compartments - however the trend between tissue types is not uniform and the authors do not say how they came up with this conclusion beyond speculation.

Reviewer #3 (Remarks to the Author):

The authors addressed my concerns in their letter; however they should have their responses incorporated into the manuscript such as a strengths and limitations section in the Discussion. They should also include the text in the Methods about what they did for combining the sequencing data across the two different capture kits to ensure the findings they reported were not due to capture kit differences.

** See Nature Research's author and referees' website at www.nature.com/authors for information about policies, services and author benefits

Point-by-point response to the reviewer comments:

Reviewer #1:

C.D. Herling and co-workers have satisfactorily to all my queries.

We would like to thank the reviewer for the positive evaluation of our manuscript.

Reviewer #2:

In their manuscript revision the authors have presented a thoroughly curated dataset and analysis of their findings but the implication of these is very limited in scope.

A few minor points.

Their comparison with external and publicly available datasets do suggest that there is an enrichment of BTG1 mutations.

Their clonal representation on Figure 2 should provide information on all mutations that delineate new clones. In its present form not all nodes are defined by distinct mutations. The authors should consider revising.

This is a good point, however, assigning all mutations to the phylogenetic trees in Figure 2 would lead to an overloaded and rather incomprehensible figure. Therefore, we added a supplementary table that contains all mutations for each identified clones of the cases shown in Figure 2.

Supplementary Figure 2

The authors postulate that methylation differences are mostly between different compartments - however the trend between tissue types is not uniform and the authors do not say how they came up with this conclusion beyond speculation.

As seen in Supplementary Table 2, purity estimates differ largely between the lymph nodes analyzed. Therefore, a uniform discrepancy of the methylation patterns within the lymph nodes is not expected.

Reviewer #3:

The authors addressed my concerns in the their letter; however they should have their responses incorporated into the manuscript such as a strengths and limitations section in the Discussion. They should also include the text in the Methods about what they did for combining the sequencing data across the two different capture kits to ensure the findings they reported were not due to capture kit differences.

Since *Nature Communications* is using a transparent peer review system, where all reviewer comments together with our responses will be published in the online material, we just added the following statement to the Discussion section: *Once large collectives of venetoclax-resistant CLL samples are available, further studies are required to elucidate these mechanisms* (page 7). In addition, we now included the description of how we homogenized data from the different enrichment kits in the Method section (page 9).